# On the Impossibility of First-Order Phase Transitions in Systems Modeled by the Full Euler Equations

**DOI:** 10.3390/e21111039

**Published:** 2019-10-25

**Authors:** Maren Hantke, Ferdinand Thein

**Affiliations:** 1Institute for Mathematics, Martin-Luther University Halle-Wittenberg, D-06099 Halle (Saale), Germany; maren.hantke@mathematik.uni-halle.de; 2Institute for Analysis and Numerics, Otto-von-Guericke University Magdeburg, PSF 4120, D-39016 Magdeburg, Germany

**Keywords:** Euler equations, phase transition, entropy principle, sharp interface, non-classical shock

## Abstract

Liquid–vapor flows exhibiting phase transition, including phase creation in single-phase flows, are of high interest in mathematics, as well as in the engineering sciences. In two preceding articles the authors showed on the one hand the capability of the isothermal Euler equations to describe such phenomena (Hantke and Thein, *arXiv*, **2017**, arXiv:1703.09431). On the other hand they proved the nonexistence of certain phase creation phenomena in flows governed by the full system of Euler equations, see Hantke and Thein, *Quart. Appl. Math.*
**2015**, *73*, 575–591. In this note, the authors close the gap for two-phase flows by showing that the two-phase flows considered are not possible when the flow is governed by the full Euler equations, together with the regular Rankine-Hugoniot conditions. The arguments rely on the fact that for (regular) fluids, the differences of the entropy and the enthalpy between the liquid and the vapor phase of a single substance have a strict sign below the critical point.

## 1. Introduction

Describing the dynamics of multiphase flows, particularly flows including phase transition, is a challenging topic in mathematics and other sciences dealing with fluid dynamics, see [1]. Furthermore, such flows appear everywhere in nature, and are of great importance in industrial applications. Usually for the modeling of phase transitions equations derived using averaging or homogenization techniques are often used—see, for instance, Zein et al. [2]. For these models, a large number of equations is used, with one set of balances for each phase or component. Due to the presence of additional differential terms, the systems are not in divergence or in a conservative form. Furthermore, exact expressions for the transfer terms are usually unknown. In fact, there is a lack of theory for these models, and their numerical solutions require a big numerical effort. On the other hand, one can use only one set of balance equations to describe the flow. An additional kinetic relation describes the exchange of mass between the phases. This concept was introduced by Abeyaratne and Knowles [3]. In the present case, such a kinetic relation has to satisfy the second law of thermodynamics. The main advantage of this type of modeling is the smaller number of equations. Moreover, due to the explicit character of the kinetic relation, it may be possible to construct exact solutions for Riemann initial data. For example, for the system of isothermal Euler equations equipped with a kinetic relation, exact solutions for Riemann problems were constructed by Hantke et al. [4,5]. Also, existence and uniqueness of the solution was proven. These results also include single-phase flows exhibiting phase creation, such as cavitation. Another hyperbolic and conservative system that may be considered are the full Euler equations. However, as it turns out, this particular system is not able to describe two-phase flows with phase transition, since certain important physical effects are missing. In this work, we want to show what is responsible for this deficiency. This work generalizes the results of Hantke et al. [6] for two-phase flows modeled using a single set of Euler equations, together with a kinetic relation. A more detailed remark is given in Section 4. The outline is as follows. In Section 2 we briefly summarize the considered model. The key argument is given in Section 3, which is finally followed by the conclusion Section 4.

## 2. Balance Laws and Entropy Inequality

In this work we consider the full system of Euler equations, which serves as a prototype for systems of hyperbolic conservation laws. In order to clearly distinguish this system from the isothermal Euler equations, we also refer to it as the *adiabatic* Euler equations. With this, we further want to emphasize that the isothermal system is not simply a subsystem of the full Euler equations. The most notable difference between these two systems is the absence of the *heat flux*
q in the adiabatic case, [7,8].

The physical fields are assumed to depend on time $t\in R_{\geq 0}$ and space $x\in R^{n},n=1,2,3$. In regular points of the bulk phases, we have the local balances for mass, momentum, and energy. That is:
(1a)$$\begin{array}{cc} \frac{\partial }{\partial t}\rho +\nabla_{x}\cdot \left(\rho v\right) & =0, \end{array}$$
(1b)$$\begin{array}{cc} \frac{\partial }{\partial t}\left(\rho v\right)+\nabla_{x}\cdot \left(\rho v⊗v\right)+\nabla_{x}p & =0, \end{array}$$
(1c)$$\begin{array}{cc} \frac{\partial }{\partial t}E+\nabla_{x}\cdot \left(v\left(E+p\right)\right) & =0. \end{array}$$
Here, ρ,v,andE denote the *density*, the *velocity*, and the *total energy*, respectively. The total energy is related to the *specific internal energy*
*e* by $E=\rho (e+v^{2}/2)$. Furthermore, the *pressure*
*p* is given via a suited *equation of state (EOS)*, that is, p=p(ρ,e). Across discontinuities, the following conditions hold:
(2a)$$\begin{array}{cc} 〚\rho (v-W)\cdot \nu 〛 & =0, \end{array}$$
(2b)$$\begin{array}{cc} \rho \left(v-W\right)\cdot \nu 〚v〛+〚p〛\nu & =0, \end{array}$$
(2c)$$\begin{array}{cc} \rho \left(v-W\right)\cdot \nu 〚e+\frac{p}{\rho }+\frac{1}{2}{\left(v-W\right)}^{2}〛 & =0\; \end{array}$$
(2d)$$\begin{array}{cc} \rho \left(v-W\right)\cdot \nu 〚s〛 & \geq 0. \end{array}$$
The brackets denote the standard jump brackets abbreviating the difference between the one-sided limits of a quantity occupying two regions separated by a discontinuity, that is, $〚\Psi 〛=\Psi_{2}-\Psi_{1}$. With $\nu \in S^{n-1}$, we denote the vector from the n−1 dimensional unit sphere, which is normal to the discontinuity, pointing from Region One to Region Two. The quantity W is the velocity of the discontinuity, which can be a *shock*, a *contact wave*, or a *phase boundary*. With Z=−ρ(v−W)·ν, we denote the mass flux. For both quantities, we will distinguish between a classical shock wave and the phase boundary (non-classical shock).
$$\begin{array}{c} Z=\left\{\begin{array}{cc} Q,\; & \mathrm{shock}\;\mathrm{wave}\\ z,\; & \mathrm{phase}\;\mathrm{boundary} \end{array}\right.\;\mathrm{and}\;W=\left\{\begin{array}{cc} S,\; & \mathrm{shock}\;\mathrm{wave}\\ w,\; & \mathrm{phase}\;\mathrm{boundary} \end{array}\right.. \end{array}$$
In our case, the phase boundary is modeled as a sharp interface, and thus it may also be considered as an *under-compressive shock wave*, cf. [9,10]. Hence, the mass flux *z* across the phase boundary has to be specified by an additional equation. Such an equation is called a kinetic relation and has to satisfy the entropy inequality (2d)
$$\begin{array}{c} \underset{=-z}{\underset{︸}{\rho (v-w)\cdot \nu }}〚s〛\geq 0, \end{array}$$
where *s* denotes the specific entropy. Thus, there is a certain freedom on how the particular kinetic relation can be chosen. Here, we assume the mass flux across the phase boundary to be given as a general function of the entropy jump. More precisely, the inequality is satisfied if *z* is given as a (monotone) function of the form: z=f(〚s〛)>0,−〚s〛>0=0,−〚s〛=0<0,−〚s〛<0.
A possible simple choice could be the linear ansatz z=−τ〚s〛 with 0<τ∈R. Nevertheless, several nonlinear choices are possible, such as: $$\begin{array}{cc} z & =-\tau 〚s{〛}^{m},\;0<\tau \in R,\;m=2k+1,\;k\in N\\ z & =-\tau \sinh(〚s〛),\;0<\tau \in R. \end{array}$$

## 3. Contradiction Argument

Here, we solely want to consider the case of a phase boundary between the liquid and vapor phases of a single substance. We assume the vapor phase to be in Region One and the liquid phase in Region Two, respectively. Below, the critical-point phase transitions between a liquid and a vapor phase are called *first-order phase transitions* [11,12]. This means that there is a jump in the first-order derivatives of the Gibbs free energy, that is, the entropy and the specific volume:(3)$$\begin{array}{cc} {\left(\frac{\partial g}{\partial T}\right)}_{p} & =-s\;\mathrm{and}\;{\left(\frac{\partial g}{\partial p}\right)}_{T}=v. \end{array}$$
For regular substances and temperatures lower than the critical temperature, the specific entropy of the vapor phase is always larger than the specific entropy of the liquid phase of the same substance, see Figure 1. Roughly speaking, this is due to the increasing number of degrees of freedom in the vapor phase compared to the liquid phase. Moreover, the entropy can be thought of as a measure for the number of microscopic states that realize a certain macroscopic state. This number is higher for the vapor phase.

Accordingly, we have
$$\begin{array}{c} 〚s〛=s_{L}-s_{V}<0\;\Rightarrow \;z=f\left(〚s〛\right)>0\;. \end{array}$$
This implies that only evaporation processes can take place and that no further thermal equilibrium can occur given the standard jump conditions for the Euler equations at the interface. Additionally, even more restrictive insights can be obtained from the energy balance at the interface. Since we have excluded the equilibrium case z=0, we have (v−w)·ν=−z/ρ≠0. Using the mass continuity we obtain for the energy balance at the interface,

$$\begin{array}{c} 0=z〚e+\frac{p}{\rho }+\frac{1}{2}{\left(v-w\right)}^{2}〛\;\Leftrightarrow \;0=〚e+\frac{p}{\rho }〛+\frac{1}{2}〚{\left(\frac{z}{\rho }\right)}^{2}〛. \end{array}$$

Introducing the *specific enthalpy*
h=e+p/ρ and using that $\rho_{L}>\rho_{V}$ (below the critical point), we yield: (4)$$\begin{array}{c} 〚h〛=-\frac{z^{2}}{2}〚\frac{1}{\rho^{2}}〛>0. \end{array}$$
However, as for the entropy, the enthalpy in the (pure) liquid phase is always smaller than the enthalpy in the (pure) vapor phase for temperatures below the critical point. This can be obtained by considering the Gibbs free energy in equilibrium 0=dg=dh−Tds. This leads to $dh=Tds\approx T(s_{L}-s_{V})<0$. The saturation curves for different variables for water using [13] are exemplary, as shown in Figure 2.

Thus, we obtain the contradiction: (5)$$\begin{array}{c} 0>〚h〛=-\frac{z^{2}}{2}〚\frac{1}{\rho^{2}}〛>0. \end{array}$$
Hence, it is *not* possible to have a phase boundary between the two (pure) phases in the classical approach that satisfies the jump conditions at the interface.

## 4. Conclusions

This work dealt with the adiabatic Euler Equations ([Disp-formula FD1a-entropy-21-01039])–(1c), together with the classical Rankine–Hugoniot conditions ([Disp-formula FD2a-entropy-21-01039])–(2d). It is therefore important to note that the heat flux q is not present in the balances for the energy and the entropy. Furthermore, we considered the case of two adjacent phases (liquid/vapor) of a single substance, and modeled the phase boundary as a sharp interface. Thus, the phase boundary is understood as a non-classical shock, and the conditions ([Disp-formula FD2a-entropy-21-01039])–(2d) should also hold at the interface. Since phase transitions between a liquid and a vapor phase are a first-order phase transition, we have jumps in the specific entropy and the specific volume. We then showed that (first-order) phase transitions between pure phases are not possible due to the difference in the entropies and enthalpies of the phases below the critical point. This result solely depends on the conditions ([Disp-formula FD2a-entropy-21-01039])–(2d), and hence is a fundamental property of the adiabatic Euler equations. To overcome this deficiency, there are two possibilities. On the one hand, heat conduction can be taken into account, which then must also appear in energy balance (1c). This would completely change the mathematical character of the system ([Disp-formula FD1a-entropy-21-01039])–(1c). On the other hand, one can extend the jump conditions by considering interface quantities, which then leads to singular solutions and the loss of self-similarity in the case of Riemann problems— see [8,14]. Hence, we conclude that the *isothermal* Euler equations are the more suitable model for studying two-phase flows with phase transition. Precisely, the isothermal Euler equations are given by ([Disp-formula FD1a-entropy-21-01039]) and (1b), that is,
$$\begin{array}{cc} \frac{\partial }{\partial t}\rho +\nabla_{x}\cdot \left(\rho v\right) & =0,\\ \frac{\partial }{\partial t}\left(\rho v\right)+\nabla_{x}\cdot \left(\rho v⊗v\right)+\nabla_{x}p & =0, \end{array}$$
and the total energy balance (1c) determines the heat flux, that is,
(6)$$\begin{array}{cc} \; & \frac{\partial }{\partial t}E+\nabla_{x}\cdot \left(v\left(E+p\right)+q\right)=0\\ \Leftrightarrow \; & \nabla_{x}\cdot q=-\left(\frac{\partial }{\partial t}\left(\rho e+\rho \frac{v^{2}}{2}\right)+\nabla_{x}\cdot \left[\left(\rho e+\rho \frac{v^{2}}{2}+p\right)v\right]\right). \end{array}$$
Thus, in contrast to the adiabatic case, the heat flux is still present in the model, although it does not need to be treated mathematically. The most important observation is that this then results in different entropy inequalities (see [8,15]), that is,
(7)$$\begin{array}{cc} 0 & \geq \frac{\partial }{\partial t}\left(\rho \left(e-Ts\right)+\rho \frac{v^{2}}{2}\right)+\nabla_{x}\cdot \left[\left(\rho \left(e-Ts+\frac{p}{\rho }\right)+\rho \frac{v^{2}}{2}\right)v\right], \end{array}$$
(8)$$\begin{array}{cc} 0 & \geq -Z〚g+\underset{=e_{kin}}{\underset{︸}{\frac{1}{2}{\left(v-W\right)}^{2}}}〛. \end{array}$$
These reflect that the *free energy*
f=e−Ts is minimized in the equilibrium of an isothermal process, [12,15]. From (8), we may then obtain a kinetic relation depending on $〚g+e_{kin}〛$ (e.g., directly proportional) which has desired thermodynamic properties, cf. [4,16]. We want to emphasize that the previous results generalize the statements given in [6] in the case where a single set of equations, together with a kinetic relation, is used to describe the two-phase flow.

In [6], we discussed the cases of nucleation and cavitation for all models based on the Euler equations, including Baer–Nunziato-type models. There, the mass transfer due to the phase transition is modeled by the source terms. In the sharp interface case, that is, Euler equations plus kinetic relation, the present results are more general in the following sense. Firstly, it is a statement about all first-order phase transitions between pure phases, and not only phase creation. Secondly, here we directly used the Rankine–Hugoniot conditions to yield a contradiction, instead of comparing the slopes of the saturation curve with a shock curve, respectively. Thirdly, it is not restricted to water as in [6], where we have used the IAPWS-IF97 EOS [13]. The present results are valid for any given substance where entropy, enthalpy, and density of the liquid and the vapor phase are related, as in Section 3.

## Figures and Tables

**Figure 1 entropy-21-01039-f001:**
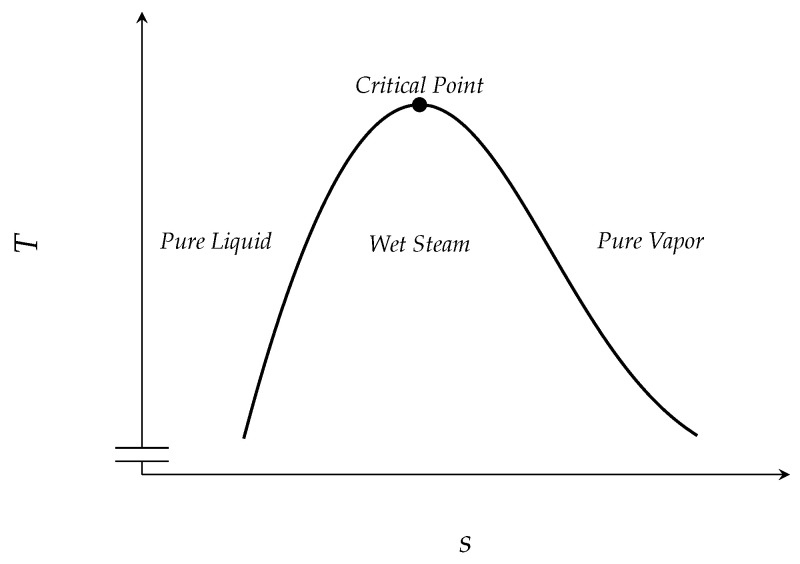
Schematic entropy–temperature diagram.

**Figure 2 entropy-21-01039-f002:**
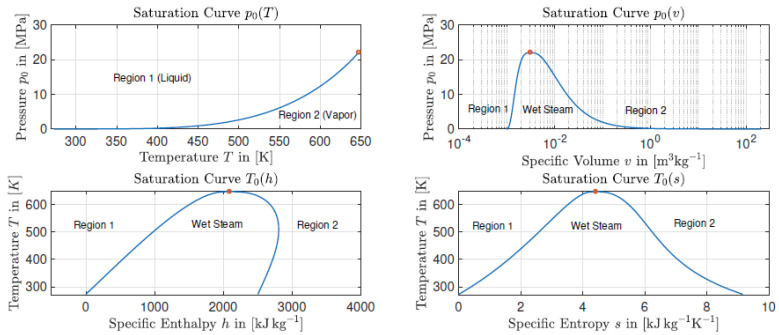
The saturation curves for different quantities calculated with the IAPWS-IF97 EOS [13].

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
