# Peer review of "On the Impossibility of First-Order Phase Transitions in Systems Modeled by the Full Euler Equations"

_entropy, 2019, doi:10.3390/e21111039_

Round 1

Reviewer 1 Report

The authors discuss in this work about the impossibility of producing the effect of condensation of water vapor to liquid water only with the models of the dynamics of the gases, by means of the system of hyperbolic equations of Euler, and the conditions of Rankine-Hugoniot when considering discontinuities. This phenomenon had already been addressed by the same authors in a 2015 work published in Quarterly of Applied Mathematics. In this paper (or rather this note), the authors try to better explain this phenomenon considering the impossibility of obtaining a two-phase flow only with Euler's equations in adiabatic state, and classical shocks.

In this sense, the work is ingenious, innovative, and in depth in this fact already observed and published by themselves four years ago.

Perhaps the only comment and criticism I could make is that the authors should present more clearly the difference of the current contribution with respect to the paper published in Q. App. Math. (2015), both from a technical-mathematical point of view and from a point view of the scope of the model. Simply clarify a little more in the introduction and conclusions, the difference between these two works. Apart from this detail, everything is perfect and I recommend that the work be published.

Author Response

We appreciate the comments of the reviewer. Concerning the remark to clearly distinguish the submitted work from the previous article, we added comments in the introduction and the conclusion. The key points are:

1) The previous work only considers nucleation and cavitation.
2) The results of the previous article are restricted to water.
3) The non-existence result from 2015 is also valid for Baer-Nunziato type models.
     In contrast, the present result discusses solely the case of a single set of Euler equations together with a kinetic relation.

Reviewer 2 Report

The authors consider in their work “On the impossibility of first order phase transitions in systems modeled by the full Euler equations” the problem of the phase transition of liquid-vapor flows of a single substance modeled by the full system of Euler equations and also related entropy inequality. The phase boundary is modeled as a sharp interface such that the mass flux across the phase boundary is modeled w.r.t the specific entropy. The main argument of this work is the contradiction in the differences of entropies and enthalpies of the phases below the critical point. The presented paper continues the work of authors, which was presented in [1, 2].  

The results and topic of presented research are scientific relevant. The paper is well written and relevant literature is cited by the authors. Therefore I can recommend this paper for publication.

Minor remarks:

2, last line: (2)_4 in place o (2)_5

Author Response

We appreciate the reviewer's effort and comments.